# Health professionals practice and associated factors towards precautionary measures for COVID-19 pandemic in public health facilities of Gamo zone, southern Ethiopia: A cross-sectional study

**Abera Mersha**[1]*, **Shitaye Shibiru**[1], **Meseret Girma**[2], **Gistane Ayele**[2], **Agegnehu Bante**[1], **Mekidim Kassa**[2], **Sintayehu Abebe**[2], **Misgun Shewangizaw**[2]

1 School of Nursing, College of Medicine and Health Sciences, Arba Minch University, Arba Minch, Ethiopia, 2 School of Public Health, College of Medicine and Health Sciences, Arba Minch University, Arba Minch, Ethiopia

* mershaabera@gmail.com

## Abstract

### Introduction

Coronavirus disease-2019 (COVID-19) is a highly contagious acute respiratory disease, which caused by a novel coronavirus. The disease disrupts health systems and resulting in social, political, and economic crises. Health professionals are in front of this pandemic and always work in a high-risk environment. The best prevention for COVID-19 is avoiding exposure to the virus. Some studies reported health professional's practice of precautionary measures for COVID-19. Nevertheless, a few have identified factors affecting. As such, this study aimed to fill those research gaps in the study setting.

### Methods

In this cross-sectional study, 428 health professionals involved from the public health facilities of the Gamo zone, southern Ethiopia. A simple random sampling method employed, and the data collected by the interviewer-administered Open Data Kit survey tool and observational checklist. The data analyzed in Stata version 15, and a binary logistic regression model used to identify factors. In this study, a statistically significant association declared at P< 0.05.

### Results

In this study, 35.3% (95%CI: 30.7%, 39.8%) of health professionals' had a good practice on precautionary measures for the COVID-19 pandemic. Use hand sanitizer or wash hands continuously with soap and water (68.9%), cover nose and mouth with a tissue during sneezing or coughing (67.3%), and use facemask in crowds (56.8%) were the most common practice reported by study participants. Marital status, being married (AOR = 1.84, 95% CI: 1.06, 3.18), good knowledge on the COVID-19 pandemic (AOR = 2.02, 95%CI: 1.02,

**Data Availability Statement:** All relevant data are within the paper and its Supporting Information files.

**Funding:** Arba Minch University provided funds for the data collection and stationary materials of this research work with a project grant code of GOV/AMU/COVID-19/PDNPI/CMHS/RCSVPO/012/2020. The website of the university is www.amu.edu.et. The funders had no role in study design, data collection, and analysis, decision to publish, or preparation of the manuscript.

**Competing interests:** The authors have declared that no competing interests exist.

3.18), and positive attitude towards precautionary measures for the COVID-19 were factors showed significant association with the practice.

## Conclusions

The magnitude of good practice of precautionary measures for the COVID-19 pandemic among health professionals was low. As such, different interventions to improve the knowledge and attitude of health professionals in the health care system are highly needed to boost the practice and to advance service delivery.

## Introduction

Coronavirus disease-2019 (COVID-19) is an emerging respiratory disease caused by a novel coronavirus or severe acute respiratory syndrome-coronavirus-2(SARS-CoV-2). The first case identified in Wuhan province, China. It is highly infectious, and clinical symptoms include fever, dry cough, fatigue, myalgia, and dyspnea [1–5]. The three modes of transmission are droplets, contact, and aerosol [6,7]. However, a few studies indicated the digestive system as a potential transmission route for COVID-19 infection [8]. The nucleic acid of SARS-CoV-2 detects in the real-time fluorescence- polymerase chain reaction (RT-PCR) [7,9,10].

The best prevention for COVID-19 is avoiding exposure to the virus [11–19]. Health care professionals (HCPs) are in front of this pandemic and always work in a high-risk environment. Preventing intra-hospital transmission of contagious disease is, therefore, a priority [20,21]. Because of their direct contact with patients, health workers play critical roles in the prevention of the COVID-19 outbreak. A combination of standards, airborne and droplet precautions should practice for all COVID-19 cases. They must use personal protective equipment (PPE) such as a glove, gown or apron, and surgical mask [16,22,23].

World Health Organization (WHO) declared the 2019-nCoV outbreak as a public health emergency of international concern (PHEIC) and as a global pandemic [24–26]. Several thousand healthcare workers have already infected with COVID-19, and there is a report of deaths in China, Italy, Spain, Pakistan, the UK, and other countries [21,27–31]. A report from China indicated that a total of 23 of the health care professionals had died in medical facilities [28].

The pandemic significantly affects the global economy. The scale of costs reduced by investment in public health systems in all economies [32]. The public health systems in Africa are coming under severe strain as the unprecedented COVID-19 pandemic persists [33,34].

Hospital associated transmission is suspected as the presumed mechanism of infection for affected health professionals (29%) and hospitalized patients (12.3%) [35]. A study conducted in Washington State reported that due to ineffective precautionary measures, 81 of the residents, 34 staff members, and 14 visitors infected and died with COVID-19 [36]. Therefore, universal source control, early identification and isolation of patients with suspected disease, the use of appropriate personal protective equipment (PPE) when caring for patients with COVID-19, and environmental disinfection are obligatory in the health care settings [37].

The practice of precautionary measures for coronavirus (CoVs) among HCPs was 89.7% in a study conducted in China [38], 87.9% in the Kingdom of Saudi Arabia [39], and 70.12% in Iran [40]. In finding from Pakistan,96.10% of HCPs had washing hands with soap/cleaning with sanitizers, and 84.30% had avoided touching of eyes, nose, or mouth to control pandemic [41], and 24.2% used facemask in the crowds in the study from Saudi Arabia [42]. Another study from the Kingdom of Saudi Arabia showed PPE used when seeing suspected cases of

CoV infection was mainly the mask (94.1%), gloves (78.8%), the gown (60%), goggles (31.8%), and the cap (22.4%) [43]. Different works of the literature indicated that age, gender, knowledge level, attitude, work experience and job category, working hours, educational attainment were factors associated with HCWs' practice of precautionary measures towards COVID-19 [38–40].

During this time, many studies are emerging by scholars regarding the COVID-19 pandemic. Nevertheless, health care professionals' practice of precautionary measures towards the COVID-19 takes the lion's share and vital to save the life of professionals and others. Presently, a few studies assessed practice and factors affecting precautionary measures for the COVID-19 pandemic among health professionals. Besides, there are limited studies in Ethiopia. Therefore, this study aimed to assess the health professional's practice and associated factors towards precautionary measures for the COVID-19 pandemic in public health facilities of the Gamo zone, southern Ethiopia.

## Materials and methods

### Study setting and period

In this study, health professionals working in public health facilities of the Gamo zone, southern Ethiopia, were involved, from June 10–19, 2020. Gamo zone is one of the administrative zones in Ethiopia. It bordered with Wolayta, Dawro, and Gofa zones in the North, on the northeast by the Lake Abaya and the southeast by the Amaro special woreda and Dirashe special woreda, and on the southwest by South Omo. The administrative center of the Gamo zone is Arba Minch town. Arba Minch town, located 505 km southwest of Addis Ababa, the capital city of Ethiopia, and 275 km southwest of Hawassa, the capital city of southern Ethiopia. Gamo zone has one administrative town and 13 woredas. It hosted five hospitals (one general and four primary hospitals), 56 health centers, and 299 health posts, which serve the community by providing preventive and curative services. There are 2570 (1096 male and 1474 female) health professionals who are working in those institutions [44].

### Study design

The institution-based cross-section study design employed to meet study objectives.

### Population

**Source population.**   The source population for this study was all health professionals who work in public health facilities of the Gamo zone, southern Ethiopia.

**Study population.**   The study population for this study was all health professionals who were working in selected health facilities of the Gamo zone, southern Ethiopia, during the study period.

### Eligibility criteria

All health professionals who were staff and working at least for six months in selected health facilities recruited in this study. Whereas those health professionals who were sick and on annual leave at the time of data collection not involved in this study.

### Sample size determination

Epi info 7 StatCalc software used to estimate the sample sizes. For the first specific objective, a single population proportion was used by considering the following assumption: P = 0.897 from the study conducted in China [38], 95% level of confidence, and 3% margin of error.

Therefore, based on this assumption, the calculated sample size was 394. To determine the sample size for the second objective two-sample comparison proportion was used by considering the following assumptions: work experience of less than one year $(P_1)$ = 82.7%, and >5 years $(P_2)$ = 95.7% from the study in Pakistan [41], 95% level of confidence, Power of 90%, and Ratio:1:1. Based on this assumption, the estimated sample size was 266. As such, the final sample size came up by adding a non-response rate of 10% to the larger sample size, which was 394 that estimated by the first objective. Therefore, the calculated sample size for this study was 434.

### Sampling procedure

Currently, there are five fully functional hospitals and fifty-six health centers in the Gamo zone. A simple random sampling method employed to select three hospitals and fifteen health centers from them (Fig 1). Then, the calculated sample size, proportionally allocated to those health facilities based on the number of health professionals who were working. Finally, a simple random sampling method after generating a table of random numbers used to select health professionals based on proportions allocated to each health facility (Table 1).

### Data collection method

Interviewer-administered Open Data Kit (ODK) survey tool and observational checklist employed to collect the data. The survey tool and observational checklist were adapted by reviewing different works of literature, WHO, and national guidelines related to the COVID-19 precautionary measures [41,45–47]. Eight data collectors and three supervisors involved in data collection.

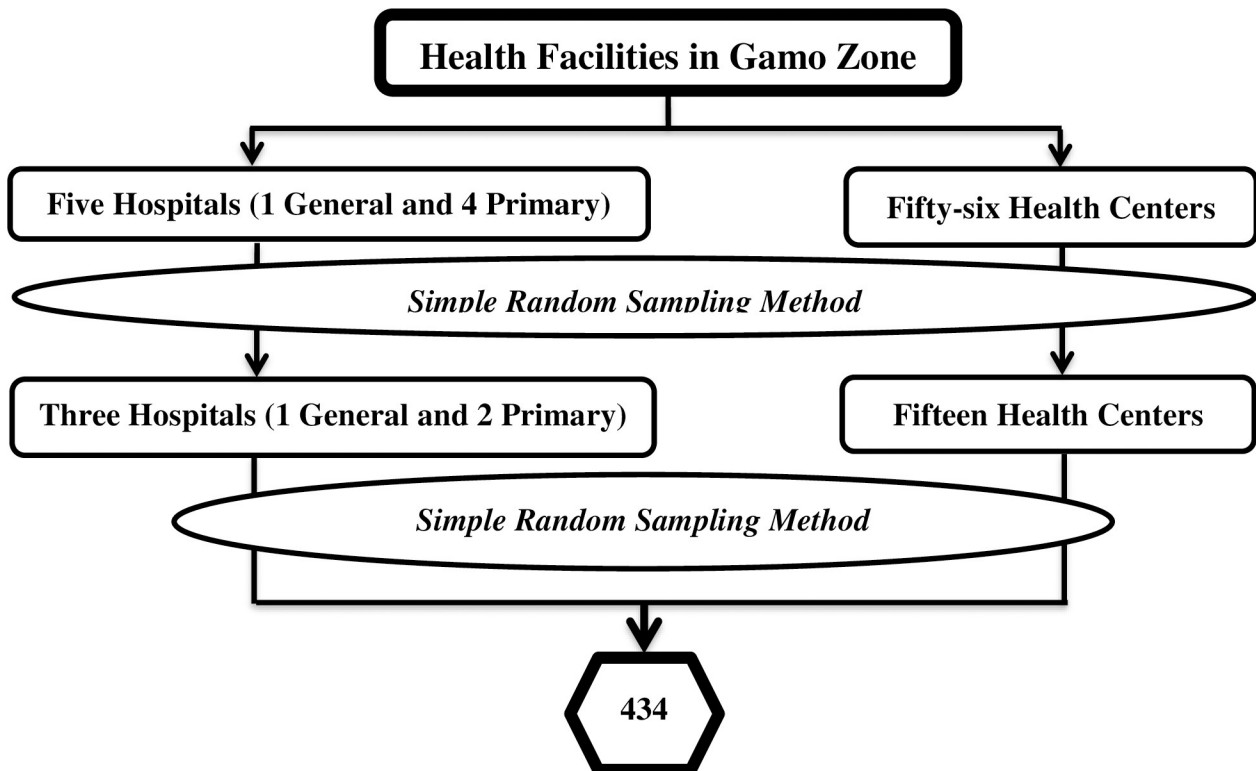

**Fig 1. Schematic presentation of the sampling procedure for the study conducted among health professionals in public health facilities of Gamo zone, southern Ethiopia, 2020.**

**Table 1. Number of the health professionals and the proportions allocated for the selected health facilities in Gamo zone, southern Ethiopia, 2020.**

| SNo | Selected health facilities | No of the health professionals | Proportion allocated (n/N*ni) |
|---|---|---|---|
| 1. | Arba Minch General Hospital | 456 | 171 |
| 2. | Gerese Primary Hospital | 62 | 23 |
| 3. | Chencha Primary Hospital | 94 | 35 |
| 4. | Sikela Health Center | 93 | 35 |
| 5. | Deramallo Health Center | 17 | 6 |
| 6. | Birbir Health Center | 54 | 20 |
| 7. | Zigiti Bakole Health Center | 16 | 6 |
| 8. | Gezeso Health Center | 27 | 10 |
| 9. | Kamba Health Center | 37 | 14 |
| 10. | Lante Health Center | 60 | 23 |
| 11. | Mengeda Health Center | 13 | 5 |
| 12. | Shelle Health Center | 29 | 11 |
| 13. | Dorze Health Center | 25 | 9 |
| 14. | Garda Health Center | 30 | 11 |
| 15. | Morka Health Center | 18 | 7 |
| 16. | Zada Health Center | 51 | 19 |
| 17. | Zefine Health Center | 59 | 22 |
| 18. | Zayse Health Center | 19 | 7 |
| Calculated sample size (n) | | | 434 |

Before starting data collection, both the data collectors and supervisors trained on data collection ways and overall procedure and ODK survey tool by experts for one day. The data collectors gave information about the study's aim before interviewing the study participants. Then those participants who were willing and signed in the voluntary informed consent interviewed. The health facilities observed for precautionary measures to the COVID-19 pandemic by incorporating different domains.

## Study variables

The dependent variable for this study was the health care professionals' practice of precautionary measures for the COVID-19. Socio-demographic and professional-related characteristics (age, sex, education level, job category, working hours, and work experience, knowledge about COVID-19, and attitude towards precautionary measures for the COVID-19 were the independent variables for this study.

## Measurements

The measurements of the outcome variable and some of the explanatory variables, stated below (Table 2).

## Data quality control

The tools validated by different scholars employed to collect the information, and it pre-tested in setting with similar characteristics. Extensive training gave to both data collectors and supervisors to maintain consistency and to standardize the data collection techniques.
The training focused on the objectives of the study, data collection tool, data collection methods, ways of checking the completeness of data, ways of maintaining the confidentiality of the data.

**Table 2. Measurements to assess the health professionals practice and associated factors towards precautionary measures for COVID-19 pandemic in public health facilities of Gamo zone, southern Ethiopia, 2020.**

| Variables | Measurements |
|---|---|
| *Health care professionals practice of precautionary measures for COVID-19* | The total score of HCPs practice of precautionary measures for COVID-19 assessment items ranged from 0–6, and a score of $\geq 4$ reported as good practice, and a score of $<4$ indicated as poor practice toward precautionary measures for COVID-19 [41]. |
| *Knowledge regarding COVID-19* | The total score of HCPs knowledge regarding COVID-19 assessment items ranged from 0–14, and a score of $\leq 10$ reported as Poor, and a score of $\geq 11$ (more than 75%) indicated a Good level [41]. |
| *Attitude towards precautionary measures for COVID-19* | The response of each item of HCPs attitude towards precautionary measures for COVID-19 recorded on a 5-point Likert scale; strongly agree (1-point), agree (2-point), undecided (3-point), disagree (4-point), and strongly disagree (5-point). Then, the total score ranges from 7 to 35, with an overall lower than mean score indicated a positive attitude toward COVID-19 [41]. |

## Data management and processing

The survey template uploaded to the ODK cloud server (ODK Aggregate platform of Arba Minch University (AMU)), then the survey template was download to the individual data collector's smartphone for data collection. After completing the data of each study participant, the data collectors sent the data to the ODK cloud server after confirmed by supervisors. After cessation of the data collection, the data downloaded from ODK aggregate and then exported to Stata version 15 for analysis.

## Data analysis

The univariate analysis, such as; proportions, frequency, and summary statistics computed. The bivariate analysis used to see the association between each independent variable and the outcome variable by using binary logistic regression. The assumptions for binary logistic regression checked, and the goodness of fit-tested by the log-likelihood ratio (LR). All variables with $P<0.25$ in the bivariate analysis included in the final model to control all possible confounders. A Multi-collinearity test ran to see the correlation between independent variables by using collinearity statistics, and Variance inflation factor (VIF) $>10$ and tolerance (T) $<0.1$ considered as suggestive of the existence of multi-collinearity. A crude and adjusted odds ratio (OR) with 95% CI estimated to identify factors affecting HCPs practice of precautionary measures for COVID-19. In this study, P-value $<0.05$ considered in declaring a result as statistically significant.

## Ethics approval and consent to participate

Ethical clearance obtained from Arba Minch University, College of Medicine and Health Sciences, Institutional Research Ethics Review Board (IRB) with reference number: IRB/408/12. Written and signed voluntary informed consent obtained from all study participants before recruiting the health professionals into the study. The concealment of the participants kept via the use of codes. The respondents also informed that the information obtained from them kept with the utmost confidentiality.

## Results

### Socio-demographic and professional related characteristics

In this study, 428 health professionals were involved, which gave a response rate of 98.6%. The mean and standard deviation of the age of study participants was 33.2±8.2 years old. Of the

participants, 240 (56.1%) were male, and 215 (50.2%) had an educational level of Diploma. One hundred thirty-one (30.6%) of participants have health professional qualifications of nurses, and 270 (63.1%) had work experience of six or more years (Table 3).

## Source of information about COVID-19 pandemic

Of the study participants, 256 (59.8%) heard information about COVID-19 from radio and television, and 207 (48.4%) from social media (Facebook, telegram, and Instagram) (Fig 2).

## Knowledge regarding COVID-19 pandemic

Three hundred ninety (91.1%) of the study participants stated that COVID-19 patients develop severe acute respiratory illness, 412 (96.3%) reported washing hands vigorously with soap and

**Table 3. Socio-demographic and professional characteristics of study participants in public health facilities of Gamo zone, southern Ethiopia, 2020 (*n* = 428).**

| Variables | Frequency | Percentage (%) |
|---|---|---|
| **Age of the participant (in a year)** | | |
| ≤30 | 221 | 51.6 |
| 31–39 | 117 | 27.3 |
| 40–49 | 60 | 14.0 |
| ≥50 | 30 | 7.0 |
| **Marital status** | | |
| Married | 291 | 68.0 |
| Other* | 137 | 32.0 |
| **Educational level** | | |
| Diploma | 215 | 50.2 |
| BSc | 154 | 36.0 |
| MSc/MPH | 13 | 3.0 |
| GP | 28 | 6.5 |
| Specialist | 18 | 4.2 |
| **Job category/profession** | | |
| Nurse | 131 | 30.6 |
| Public health | 47 | 11.0 |
| Midwives | 78 | 18.2 |
| Pharmacy | 31 | 7.2 |
| Lab technician | 43 | 10.0 |
| Physician | 45 | 10.5 |
| Other® | 53 | 12.4 |
| **Work experience (in a year)** | | |
| 1–3 | 104 | 24.3 |
| 4–5 | 54 | 12.6 |
| ≥6 | 270 | 63.1 |
| **Working hours per day (in hr.)** | | |
| <8 | 11 | 2.6 |
| ≥8 | 417 | 97.4 |

*Single, divorced and widowed

®Environmental health, IESO, Anesthesia, Radiology, Dentist, and Psychiatry.

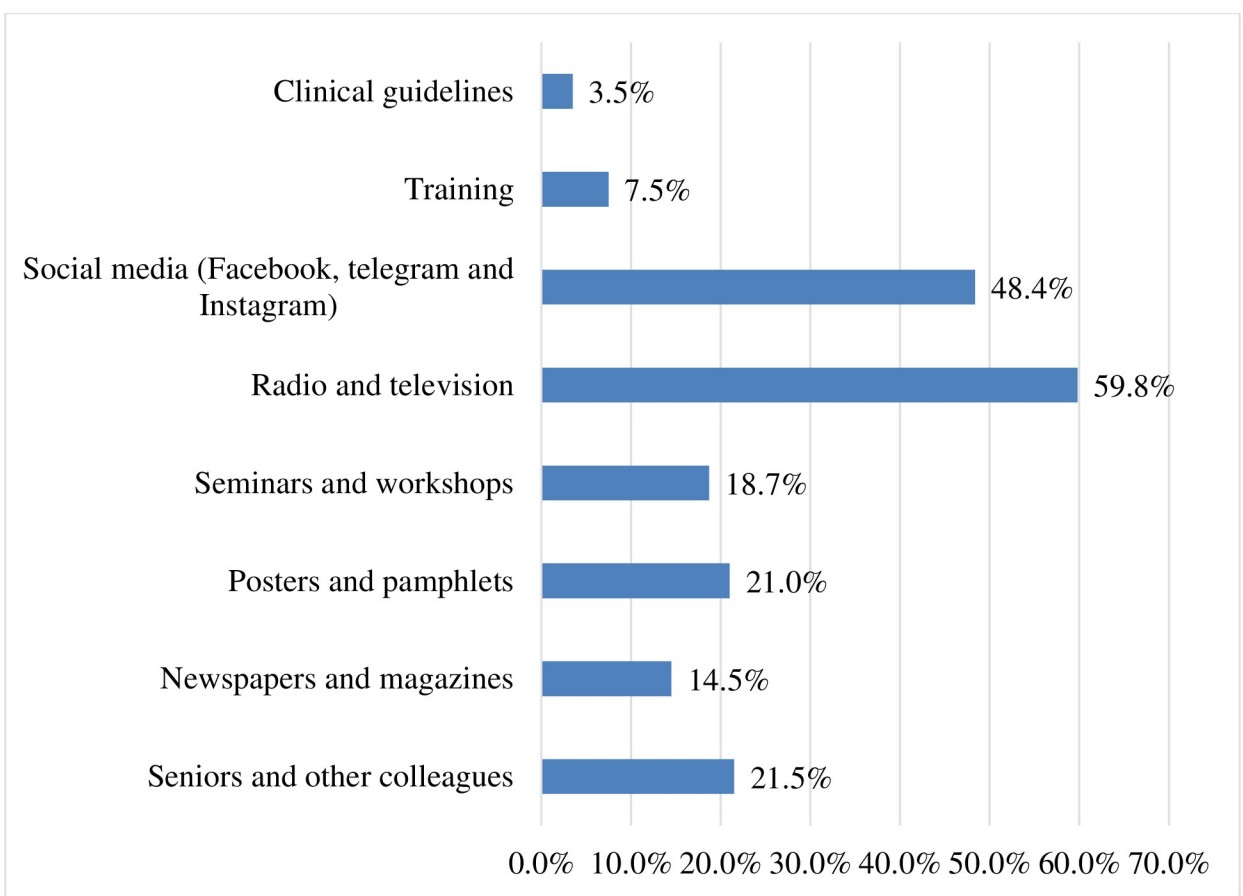

**Fig 2. Source of information about COVID-19 pandemic for health professionals in public health facilities of Gamo zone, southern Ethiopia, 2020 (*n = 428*).**

water can prevent COVID-19, and 403 (94.2%) said fever, cough, and shortness of breath are symptoms for COVID-19 (Fig 3). Overall, 84.1% (95%CI: 80.6%, 87.6%) of health professionals had good knowledge about the COVID-19 pandemic.

## Attitude towards precautionary measures for COVID-19 pandemic

Out of study participants, 309 (72.2%) strongly agreed that gowns, gloves, masks, and goggles must use when dealing with the COVID-19 patients, and 161 (37.6%) agreed that healthcare workers must acknowledge themselves with all the information about the COVID-19 (Table 4). Overall, 53% (95%CI: 48.3%, 57.8%) of HCPs had a positive attitude towards precautionary measures for the COVID-19 pandemic.

## Observational findings regarding the practice of precautionary measures for COVID-19 pandemic at the facility level

Of the health facilities, in eight (44.4%), a team of HCWs should always bed signaled to care exclusively for suspected or confirmed cases to reduce the risk of transmission. In fourteen (77.8%) health facilities, all persons entering the patients' room never recorded (Table 5).

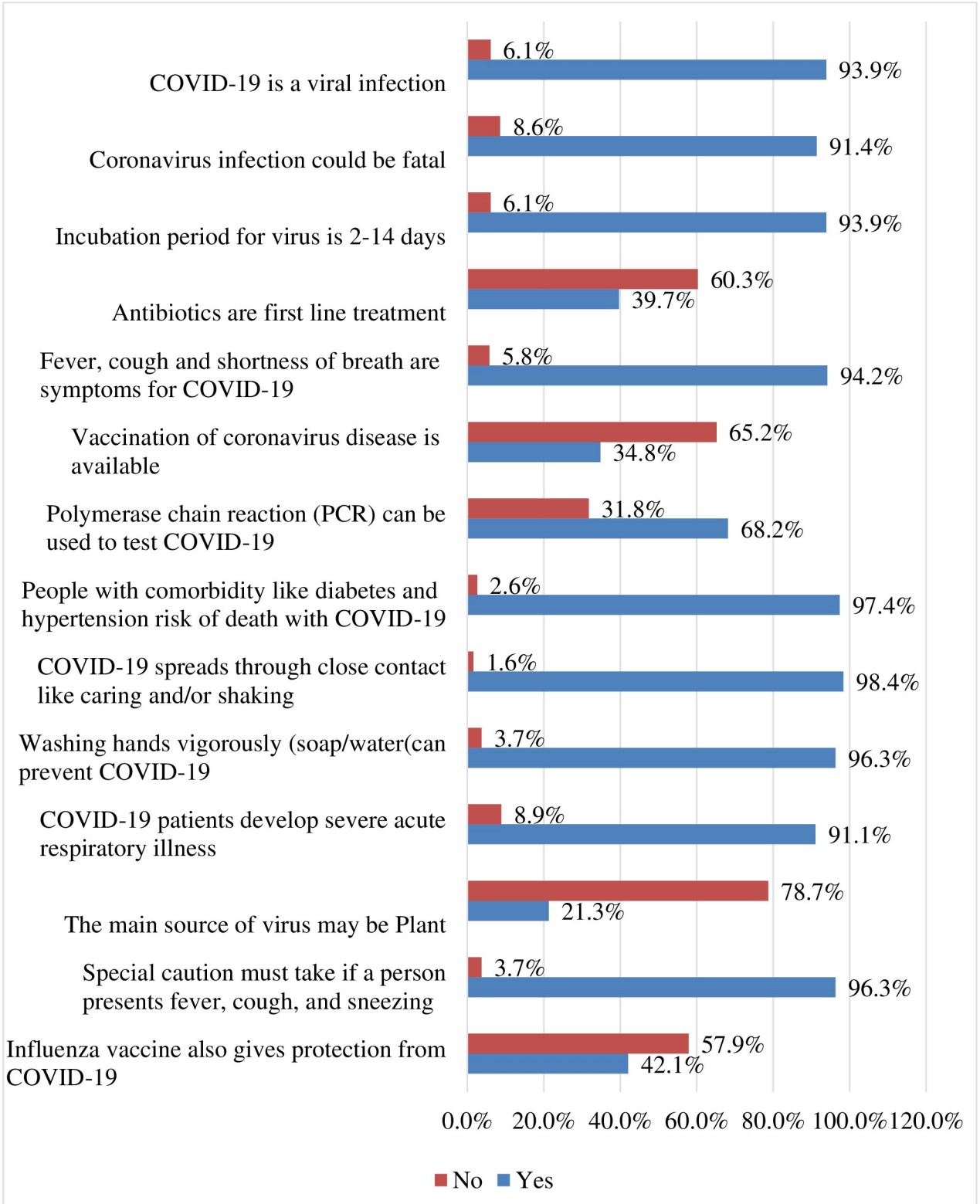

**Fig 3. Knowledge of health professionals about COVID-19 pandemic for health professionals in public health facilities of Gamo zone, southern Ethiopia, 2020 (*n = 428*).**

**Table 4. Attitude of health professionals towards precautionary measures for COVID-19 pandemic in public health facilities of Gamo zone, southern Ethiopia, 2020 (n = 428).**

| Characteristics | Strongly disagree | Disagree | Neutral | Agree | Strongly agree |
|---|---|---|---|---|---|
| | N (%) | N (%) | N (%) | N (%) | N (%) |
| Healthcare workers must acknowledge themselves with all the information about COVID-19 | 3(0.7) | 27(6.3) | 49 (11.4) | 161 (37.6) | 188(43.9) |
| Transmission of COVID-19 infection can prevent by using universal precautions given by WHO, CDC | 7(1.6) | 7(1.6) | 35(8.2) | 142 (33.2) | 237(55.4) |
| Any related information about COVID-19 should disseminate among healthcare workers | 13(3.0) | 27(6.3) | 25(5.8) | 157 (36.7) | 206(48.1) |
| Prevalence of COVID-19 can reduce by the active participation of healthcare workers in the hospital infection control program | 3(0.7) | 8(1.9) | 32(7.5) | 146 (34.1) | 239(55.8) |
| Intensive and emergency treatment should give to diagnosed patients | 15(3.5) | 41(9.6) | 26(6.1) | 130 (30.4) | 216(50.5) |
| COVID-19 patients should keep in isolation | 4(0.9) | 5(1.2) | 15(3.5) | 105 (24.5) | 299(69.9) |
| Gowns, gloves, masks, and goggles must use when dealing with COVID-19 patients | 1(0.2) | 2(0.5) | 10(2.3) | 106 (24.8) | 309(72.2) |

## Health professionals practice on precautionary measures for COVID-19 pandemic

Of the study participants, 295 (68.9%) wash hands continuously with water and soap or use hand sanitizer, and 220 (51.4%) sometimes educate their patients about the COVID-19 (Fig 4). Overall, 35.3% (95%CI: 30.7%, 39.8%) of health professionals had a good practice on precautionary measures for the COVID-19 pandemic.

## Factors associated with the health professionals practice of the precautionary measures for the COVID-19 pandemic

In a multivariable model, marital status (being married), good knowledge about COVID-19, and a positive attitude towards precautionary measures had shown a significant association with health professionals' practice of precautionary measures for the COVID-19 pandemic.

Married health professionals were 1.84 times more likely to practice precautionary measures for COVID-19 in the health facility as compared to counterparts (AOR = 1.84, 95%CI: 1.06, 3.18). The odds of good practice of precautionary measures for the COVID-19 pandemic were 2.02 in health professionals with good knowledge of COVID-19 (AOR = 2.02, 95%CI: 1.02, 3.99). Health professionals with a positive attitude were 29% more likely to practice precautionary measures for the COVID-19 pandemic in a public health facility (AOR = 3.29, CI: 2.09, 5.19) (Table 6).

## Discussion

This survey aimed to fill a research gap in Ethiopia by assessing health professionals' practice of precautionary measures for COVID-19 in public health facilities. In this study, 2/3[rd] of health professionals' had knowledgeable about the COVID-19 pandemic, 1/2[nd] were a positive attitude towards precautionary measures for COVID-19, and only 1/3[rd] had a good practice on precautionary measures in public health facilities. Marital status, knowledge about COVID-19, and attitude towards precautionary measures were factors identified in this study that showed significant association with good practice of precautionary measures for the COVID-19 pandemic.

**Table 5. Observational findings of the practice of precautionary measures for COVID-19 pandemic in public health facilities of Gamo zone, southern Ethiopia, 2020 (_n_ = 18).**

| Characteristics | Never | Sometimes | Always |
|---|---|---|---|
| | N (%) | N (%) | N (%) |
| Hand hygiene stations and waste bins installed at strategic locations across the health facility | 6(33.3) | 9(50.0) | 3(16.7) |
| Health care workers applying standard precautions for all patients | 8(44.4) | 7(38.9) | 3(16.7) |
| Droplets and contact precautions recommended | 5(27.8) | 10(55.6) | 3(16.7) |
| Patients placed in the adequately ventilated rooms | 5(27.8) | 7(38.9) | 6(33.3) |
| A one-meter distance between beds maintained | 4(22.2) | 6(33.3) | 8(44.4) |
| Equipment is either single-use or disposable or if equipment (e.g., stethoscopes, blood pressure cuffs, thermometers, food trays) needs to be shared among patients, clean and disinfect between use for each patient (e.g., by using ethyl alcohol 70%) | 9(50.0) | 8(44.4) | 1(5.6) |
| Routinely clean and disinfect surfaces with which the patient is in contact | 8(44.4) | 8(44.4) | 2(11.1) |
| Health care worker apply droplet and contact precautions before entering the room | 10 (55.6) | 6(33.3) | 2(11.1) |
| Health care workers are used airborne precautions for aerosol-generating procedures | 7(38.9) | 7(38.9) | 4(22.2) |
| Team of HCWs should bed signaled to care exclusively for suspected or confirmed cases to reduce the risk of transmission | 4(22.2) | 6(33.3) | 8(44.4) |
| Staffs receive training on standard, contact, droplets, and airborne precautions | 5(27.8) | 7(38.9) | 6(33.3) |
| Adequate personal protective equipment (PPE) is easily accessible to staff | 6(33.3) | 8(44.4) | 4(22.2) |
| Avoid moving and transporting patients out of their room or area unless medically necessary | 9(50.0) | 3(16.7) | 6(33.3) |
| HCWs who are transporting patients perform hand hygiene and wear appropriate PPE. | 4(22.2) | 10(55.6) | 4(22.2) |
| The area receiving the patient arrange for all necessary precautions as early as possible before the patient's arrival | 8(44.4) | 7(38.9) | 3(16.7) |
| Visitors essential for patient support are limited | 11 (61.1) | 4(22.2) | 3(16.7) |
| Visitors apply droplet and contact precautions | 9(50.0) | 5(27.8) | 4(22.2) |
| All persons entering the patient's room recorded | 14 (77.8) | 4(22.2) | 0(0.0) |
| Manage laboratory specimens, laundry, food service utensils, and medical waste following safe routine procedures according to Infection Prevention and Control (IPC) guidelines. | 7(38.9) | 8(44.4) | 3(16.7) |

In this study, the magnitude of good practice of precautionary measures for COVID-19 was 35.3% (95%CI: 30.7%, 39.8%). These were very low as compared to studies conducted in Iran (70.12%) [40], in China (89.7%) [38], the Kingdom of Saudi Arabia (87.9%) [39], Uganda (74%) [48], and Ethiopia (62%) [49]. The finding of this study indicated that 50.5% of health professionals avoided touching mouse, eyes, and noses, and 68.9% had wash hands continuously with water, soap, or use hand sanitizer. These were incongruent with studies conducted in Pakistan (84.3% for avoiding touching eyes, nose, and mouth, and 96.1% for hand washing and use sanitizer) [41], in two studies from Saudi Arabia, 24.2% [42], and 94.1% [43] use facemask in crowds. The observational finding of this study also indicated that a gap in applying droplet and contact precautions (using PPEs), routinely cleaning and disinfecting surfaces with which the patient contact and equipment used, and controlling visitors (there is overcrowding) in the health facility. This discrepancy may be due to differences in socio-demographic and economic characteristics, technological advancement and health care system, and social-cultural factors. The other factor may be methodological aspects (the majority of previous studies were an online survey (using email, Facebook, and telegram) that may overestimate the practice.

The health professionals' marital status significantly associated with good practice of precautionary measures for COVID-19. However, other socio-demographic and professional factors (sex, age, educational attainment, job category/profession, and work experience) had not shown significant association. These were not in line with studies conducted in China [38], the Kingdom of Saudi Arabia [39], Iran [40], Pakistan [41], and Uganda [48]. This difference might related to sampling variation or sample clustering that previous studies mainly based on

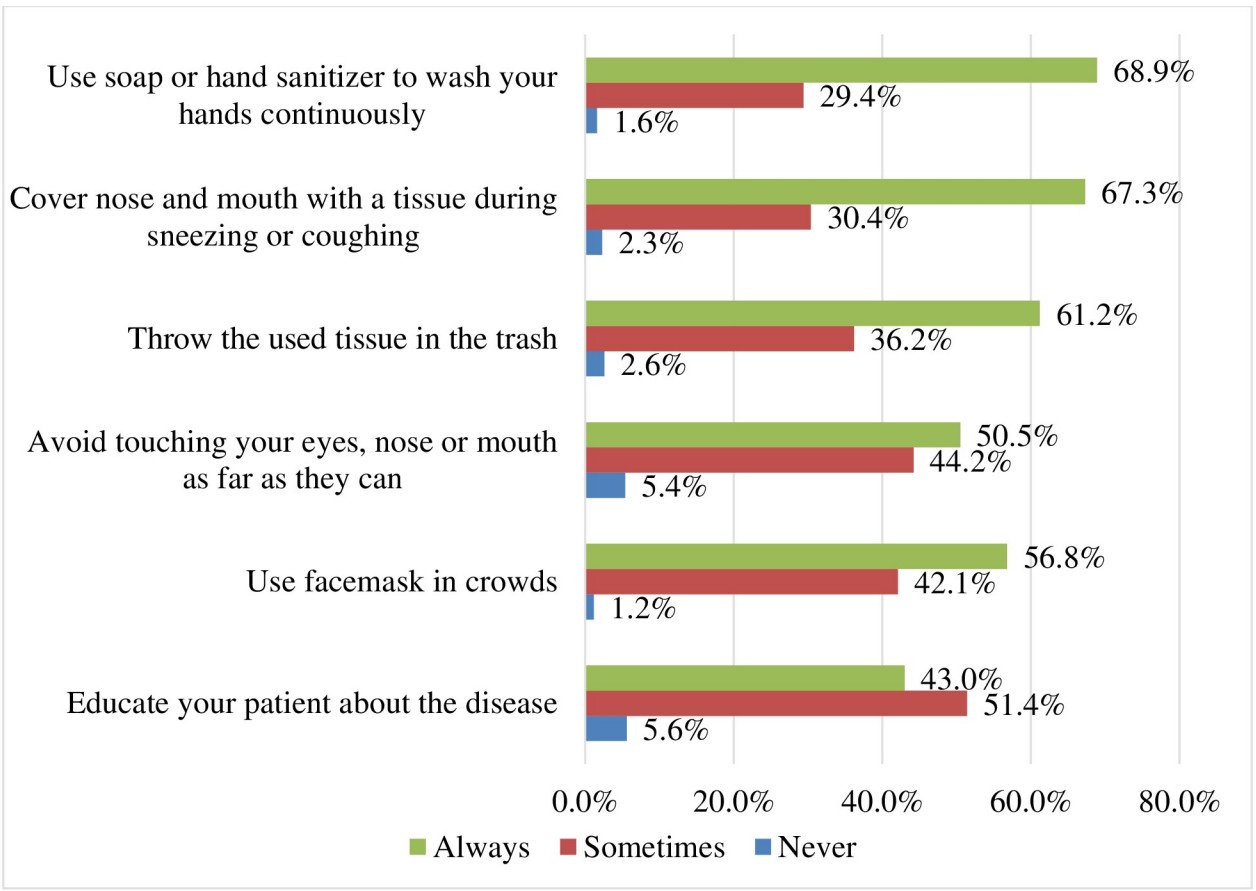

**Fig 4. Health professionals practice of precautionary measures for COVID-19 pandemic in public health facilities of Gamo zone, southern Ethiopia, 2020 (*n* = 428).**

an online survey, and participants are only those who access the internet service during the data collection period. Besides, a health professional who continuously manages patients with COVID-19 might not involve in that study due to time shortage. Coronavirus disease-2019 is a global pandemic that all the information sources (mass media, stream media, social media, and others) and all the government sectors give focus on this issue, and health professionals seek information utmost an equal level. Therefore, the practice of precautionary measures for COVID-19 in health facilities does not that much vary with sex, age category, educational level, job category, and work experience difference.

In this study, knowledge of health professionals' on the COVID-19 had shown a significant association with a good practice of precautionary measures in health facilities. These were in line with studies conducted in Pakistan [41], Iran [40], the Kingdom of Saudi Arabia [39], China [38], Ethiopia [49,50], and Uganda [48]. Similarly, the attitude of the health profession-als' towards precautionary measures had shown a significant association with practice in the health facility. These were in line with the study conducted in Pakistan [41]. These are facts that health professionals who are knowledgeable about the pandemic and positive attitude towards precautionary measures are more likely to put in practice. In general, the positive atti-tude of the health professional is a base to seeking information about the pandemic from dif-ferent sources and concerning bodies to build knowledge and resulted in behavioral change and to put the practice in the ground.

**Table 6. Bi-variable and multivariable analysis of factors associated with health professional practice of precautionary measures for COVID-19 pandemic in public health facilities of Gamo zone, southern Ethiopia, 2020 (*n* = 428).**

| Variables | Practice | | Crude OR | Adjusted OR | P-value |
|---|---|---|---|---|---|
| | **Good** | **Poor** | **95%CI** | | |
| **Sex of the participant** | | | | | |
| Male | 76(50.3%) | 164(59.2%) | 1 | 1 | |
| Female | 75(49.7%) | 113(40.8%) | 1.43(0.96,2.13) | 1.41(0.89,2.22) | 0.14 |
| **Age (in a year)** | | | | | |
| ≤30 | 75(49.7%) | 146(52.7%) | 1 | 1 | |
| 31–39 | 45(29.8%) | 72(26.0%) | 1.22(0.76,1.94) | 1.11(0.63,1.93) | 0.72 |
| 40–49 | 20(13.2%) | 40(14.4%) | 0.97(0.53,1.78) | 0.72(0.35,1.49) | 0.38 |
| ≥50 | 11(7.3%) | 19(6.9%) | 1.13(0.51,2.49) | 1.14(0.44,2.96) | 0.79 |
| **Marital status** | | | | | |
| Married | 116(76.8%) | 175(63.2%) | 1.93(1.23,3.03) | 1.84(1.06,3.18)* | **0.03** |
| Other® | 35(23.2%) | 102(36.8%) | 1 | 1 | |
| **Educational level** | | | | | |
| Diploma | 79(52.3%) | 136(49.1%) | 1.14(0.77,1.69) | 0.91(0.54,1.56) | 0.74 |
| Other± | 72(47.7%) | 141(50.9%) | 1 | 1 | |
| **Job category/Profession** | | | | | |
| Nurse | 55(36.4%) | 76(27.4%) | 1.83(0.92,3.66) | 1.57(0.71,3.47) | 0.27 |
| Public health | 15(9.9%) | 32(11.6%) | 1.19(0.51,2.79) | 0.87(0.34,2.23) | 0.77 |
| Midwives | 27(17.9%) | 51(18.4%) | 1.34(0.63,2.86) | 1.33(0.57,3.12) | 0.51 |
| Pharmacy | 10(6.6%) | 21(7.6%) | 1.21(0.46,3.16) | 1.17(0.41,3.36) | 0.77 |
| Lab technician | 16(10.6%) | 27(9.7%) | 1.50(0.64,3.55) | 1.89(0.72,4.97) | 0.19 |
| Physician | 13(8.6%) | 32(11.6%) | 1.03(0.43,2.48) | 0.87(0.34,2.25) | 0.77 |
| Other© | 15(9.9%) | 38(13.7%) | 1 | 1 | |
| **Work experience(in a year)** | | | | | |
| 1–3 | 28(18.5%) | 76(27.4%) | 1 | 1 | |
| 4–5 | 15(9.9%) | 39(14.1%) | 1.04(0.50,2.18) | 0.86(0.39,1.89) | 0.70 |
| ≥6 | 108(71.5%) | 162(58.5%) | 1.81(1.10,2.98) | 1.33(0.70,2.54) | 0.38 |
| **Knowledge about COVID-19** | | | | | |
| Poor | 15(9.9%) | 53(19.1%) | 1 | 1 | |
| Good | 136(90.1%) | 224(80.9%) | 2.15(1.16,3.95) | 2.02(1.02,3.99)* | **0.04** |
| **Attitude towards precautionary measures** | | | | | |
| Negative | 45(29.8%) | 156(56.3%) | 1 | 1 | |
| Positive | 106(70.2%) | 121(43.7%) | 3.04(1.99,4.63) | 3.29(2.09,5.19)* | **<0.001** |

®single, widowed and divorced, ±BSc, MSc, MPH, GP, and Specialist

© Environmental health, IESO, Anesthesia, Radiology, Dentist, and Psychiatry, and

*Significant at P-value<0.05.

The main strength of this study was that it assessed the health professionals' practice and factors affecting precautionary measures for the COVID-19 pandemic in the health facilities with limited previous studies. It also used validated Open Data Kit survey tools to collect the information.

The limitation of this study was the study might subject to recall and social desirability biases. The study, conducted in a very constrained environment that different activities controlled by the national emergency team due to the pandemic. The study was cross-sectional, which the causal relationship was under caution. Therefore, those issues must consider while interpreting the study findings.

The implication of this study is; it involved health professionals in health facilities, and health professionals are in the front line to this pandemic, and different interventions needed in the ground route to improve service quality and to stable the health care system. Currently, millions of individuals are infected, and thousands died from this evil disease. The pandemic disrupts the health care system. Therefore, assessing the practice and identifying the factors affect is very important to intervene urgently to squeal the consequence within a short period. The finding of this study can urge different stakeholders, task forces, and public health emergency teams to design strategies for intervention.

In summary, there are limited studies that showed the health professionals' practice and factors affecting precautionary measures for the COVID-19 pandemic. Therefore, this study aimed to fill these research gaps in Ethiopia. This study identified that there was a gap in the practice of precautionary measures for COVID-19 among health professionals, and knowledge and attitude towards precautionary measures were the most determinate factors. The observational finding also supplementary for the results indicated based on interviewed data. This study had its limitation, and the results must interpret by considering those limitations. These results of this study can be input for health facilities in the Gamo zone, different stakeholders, and task forces to design specific strategies for intervention.

## Conclusions

This study showed that the practice of precautionary measures for the COVID-19 pandemic was low. Even if, majority of the health professionals know about COVID-19, there was a gap. Marital status, knowledge about COVID-19, and attitude towards the precautionary measures identified as factors. Therefore, the investigators recommended that the ministry of health and other concerning task forces should provide capacity-building activities such as in-service training, motivate and recognize staffs to improve the knowledge, and to change the behavior or attitude of the health professionals.

## Supporting information

**S1 File. English version tool and observational checklist.**
(PDF)

## Acknowledgments

Our heartfelt thanks go to the Gamo zone health department, Chief executive officers in the hospitals, health center heads, task force leaders who gave support during data collection, data collectors, and study participants. Finally, yet importantly, we would like to say thank you for all peoples who support directly or indirectly.

## Author Contributions

**Conceptualization:** Abera Mersha, Shitaye Shibiru, Meseret Girma, Agegnehu Bante, Sintayehu Abebe.

**Data curation:** Abera Mersha, Shitaye Shibiru, Meseret Girma, Gistane Ayele, Agegnehu Bante, Sintayehu Abebe.

**Formal analysis:** Abera Mersha, Shitaye Shibiru, Mekidim Kassa.

**Funding acquisition:** Abera Mersha, Shitaye Shibiru, Agegnehu Bante, Mekidim Kassa, Misgun Shewangizaw.

**Investigation:** Abera Mersha, Shitaye Shibiru, Gistane Ayele, Sintayehu Abebe.

**Methodology:** Abera Mersha, Meseret Girma, Gistane Ayele, Agegnehu Bante, Mekidim Kassa.

**Project administration:** Abera Mersha.

**Resources:** Abera Mersha, Misgun Shewangizaw.

**Software:** Abera Mersha, Shitaye Shibiru.

**Supervision:** Abera Mersha, Shitaye Shibiru, Gistane Ayele, Agegnehu Bante.

**Validation:** Abera Mersha.

**Visualization:** Abera Mersha, Shitaye Shibiru.

**Writing – original draft:** Abera Mersha.

**Writing – review & editing:** Abera Mersha.

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
