## [Decision Letter · Decision Letter 0]

24 Nov 2020

PONE-D-20-27501

Health professionals practice and associated factors towards precautionary measures for COVID-19 pandemic in public health facilities of Gamo zone, southern Ethiopia: a cross-sectional study

PLOS ONE

Dear Dr. Mersha,

Thank you for submitting your manuscript to PLOS ONE. After careful consideration, we feel that it has merit but does not fully meet PLOS ONE’s publication criteria as it currently stands. Therefore, we invite you to submit a revised version of the manuscript that addresses the points raised during the review process.

Please read carefully the comments provided below.

We look forward to receiving your revised manuscript.

Kind regards,

Marzia Lazzerini, PhD

Academic Editor

PLOS ONE

Additional Editor Comments:

The paper has been already published as a pre-print https://www.medrxiv.org/content/10.1101/2020.09.05.20188805v1.full.pdf), please check Plos One policies on preprints (https://journals.plos.org/plosone/s/submit-now) and refer to the editor to additional information.

In terms of content the paper brings relevant data on COVID prevention in setting with low resources. However, in order to reconsider the paper for publication please consider the following aspects 

1) refer to the STROBE guideline for reporting of cross sectional studies https://journals.plos.org/plosone/s/submission-guidelines#loc-guidelines-for-specific-study-types

2) improve use of language (grammar and syntax) consider revision from an external 

3) make sure that the paper is submitted according to the Plos One guideline of reporting https://journals.plos.org/plosone/s/submission-guidelines.

Please read carefully the comments from the referee and revise accordingly.

Journal Requirements:

2. Please provide additional details regarding participant consent. In the ethics statement in the Methods and online submission information, please clarify if consent was informed consent.

3. Please provide an additional copy of your questionnaire in the original language as an additional supplementary file.

5. Thank you for submitting the above manuscript to PLOS ONE. During our internal evaluation of the manuscript, we found significant text overlap between your submission and the following previously published works, some of which you may be an author.

https://www.nejm.org/doi/full/10.1056/NEJMc2005696?af=R&rss=currentIssue

https://link.springer.com/article/10.1186/s40249-020-00646-x?code=e16360c7-f87d-436f-8d78-19dfd233c878&error=cookies_not_supported

https://www.liverpoolchampion.com.au/story/6660523/pandemic-could-knock-23-trillion-or-more-from-global-economy/?cs=17267

https://www.researchsquare.com/article/rs-30341/v1

https://www.medrxiv.org/content/10.1101/2020.04.13.20063198v1.full.pdf

Please revise the manuscript to rephrase the duplicated text, cite your sources, and provide details as to how the current manuscript advances on previous work. Please note that further consideration is dependent on the submission of a manuscript that addresses these concerns about the overlap in text with published work.

Reviewers' comments:

Reviewer's Responses to Questions

**Comments to the Author**

1. Is the manuscript technically sound, and do the data support the conclusions?

Reviewer #1: Partly

Reviewer #2: Partly

2. Has the statistical analysis been performed appropriately and rigorously? 

Reviewer #1: I Don't Know

Reviewer #2: No

3. Have the authors made all data underlying the findings in their manuscript fully available?

Reviewer #1: Yes

Reviewer #2: Yes

4. Is the manuscript presented in an intelligible fashion and written in standard English?

Reviewer #1: Yes

Reviewer #2: Yes

5. Review Comments to the Author

Reviewer #1: The journal should be happy to receive manuscripts from frontline workers trying to cope with the Covid-19 pandemic in resource poor countries. This is an example of combined operations research (studies of whether the health workers are doing the things right – according to clear guidelines) and implementation research (whether we are doing the right things – how the intentions of the guidelines are translated into the right actions). That the study is executed in close collaboration between the local university and the active health services in Goma Zone is also highly commendable. Unfortunately, these two are a bit confused in this manuscript. What are named the first and second objective (good practice / associated factors) do not clearly refer to a specific guideline and factors associated or not to good practice have a less than obvious link to practice. E.g. the questions used to map knowledge about the virus and preventive measures might not clearly lead to optimal practice, even with the best of intentions. If I am not wrong, and their indicators are not based on clear national guidelines the study did not use standardized questionnaires published elsewhere, the authors need to express this clearly in the revised manuscript. They need to describe how they developed their own indicators and questionnaire. It is indicated that observations were done, but it is difficult to see the findings.

If this paper should be consider of general interest we need to know more about the local context: the population, geography / transportation, economy, language / communication channels, education and degree of Covid-19 exposure.

A number of technical / formal details need to be corrected. The English needs a total brush-up, a thorough proof reading should be done and abbreviations are not always introduced / used correctly (ODK,AMU,IPC?).

Lastly: in implementation research the dissemination of the results to similar settings and feed-back to the study subjects are essential factors. A paragraph on this should be added. There is room for shortening and reduction of tables to make space for the text I am missing.

Reviewer #2: It appears that this has been previously published and therefore does not qualify for publication with PLOS ONE. In terms of the technical content of your paper I find the initial analysis good. However some key issues not dealt with in your paper include the failure to acknowledge that you've interviewed only healthcare professions who are likely to engage in many of the practices you identify as standard practice--even without COVID-19. You did find that only 68.9% practice some of these standard preventative measures--but there isn't a great deal of discussion as to why these attitudes exist. This reads a bit more like an accounting of healthcare workers perceptions rather than and explanation of why which would have been a bit more useful. Also your multivariate model looks like an attempt to throw everything in and see what ends up being significant. There is not theoretical framework or justification for why the independent variables were chosen and since this is not a grounded theory study I would expect the authors to use some kind of theoretical model to justify their inclusion of these variables. This also leads to a week discussion of these variables and the mechanisms through which they may be influencing the dependent variable (use of good precautionary practices).

6. PLOS authors have the option to publish the peer review history of their article (what does this mean?). If published, this will include your full peer review and any attached files.

Reviewer #1: **Yes: **Gunnar Bjune

Reviewer #2: No

---

## [Author Response · Author response to Decision Letter 0]

3 Dec 2020

Responses for Editor and Reviewers Comments

Manuscript Number: PONE-D-20-27501

Manuscript Title: Health professionals practice and associated factors towards precautionary measures for COVID-19 pandemic in public health facilities of Gamo zone, southern Ethiopia: a cross-sectional study

Dear sir/Madam

In the very beginning, I would like to say thanks to the editor and reviewers for their constructive comments and valuable suggestions. After saying that here are the changes made based on the given comments, and responses for some of the comments or suggestions: 

Editor Comments:

1. Check Plos One policies on preprints

Response: This manuscript was published as preprints in medRxiv preprint doi: https://doi.org/10.1101/2020.09.05.20188805. This information is included as additional information in the revised manuscript. 

2. Refer to the STROBE guideline for reporting of cross sectional studies

Response: I referred the STROBE guideline for reporting of cross sectional studies, and I assured that our article is in line with the guideline.

3. Improve use of language (grammar and syntax) consider revision from an external

Response: The whole manuscript is copy edited and revised for the grammar and syntax.

4. Make sure that the paper is submitted according to the Plos One guideline

Response: I assured that our paper is submitted according to the Plos One guideline

Additional requirements:

1. Please ensure that your manuscript meets PLOS ONE's style requirements, including those for file naming

Response: I assured that our paper meets PLOS ONE's style requirements, and file naming

2. Please provide additional details regarding participant consent. In the ethics statement in the Methods and online submission information, please clarify if consent was informed consent.

Response: The section modified and re-stated as “informed consent” in the revised manuscript. 

3. Please provide an additional copy of your questionnaire in the original language as an additional supplementary file.

Response: The questionnaire was original prepared in English and it used as it is, because the study participants are health professionals. It included as Supporting Information

“S1 English Version Tool and Observational Checklist” in the main manuscript, and the document was uploaded as separate file. 

4. To change financial disclosure

Response: The funding statement or financial disclosure is improved and the revised statement is included at the end of cover. “See cover letter”

5. Thank you for submitting the above manuscript to PLOS ONE. During our internal evaluation of the manuscript, we found significant text overlap between your submission and the following previously published works, some of which you may be an author.

Response: I paraphrased the sections with significant text overlap, cited in the text, and indicated the source as much as possible.

Reviewers Comments:

Reviewer #1

1. This is an example of combined operations research (studies of whether the health workers are doing the things right – according to clear guidelines) and implementation research (whether we are doing the right things – how the intentions of the guidelines are translated into the right actions). That the study is executed in close collaboration between the local university and the active health services in Goma Zone is also highly commendable. Unfortunately, these two are a bit confused in this manuscript. What are named the first and second objective (good practice / associated factors) do not clearly refer to a specific guideline and factors associated or not to good practice have a less than obvious link to practice. E.g. the questions used to map knowledge about the virus and preventive measures might not clearly lead to optimal practice, even with the best of intentions. If I am not wrong, and their indicators are not based on clear national guidelines the study did not use standardized questionnaires published elsewhere, the authors need to express this clearly in the revised manuscript. They need to describe how they developed their own indicators and questionnaire.

Response: I thanks for this comment. As you stated, this article is combined operational and implementation research. The survey tool and observational checklist were adapted by reviewing different works of literature that published, WHO, and national guidelines related to COVID-19 precautionary measures during that time as indicated and cited, in the data collection method. Therefore, the indicators and questionnaire that used in this study validated and used by different scholars. Nevertheless, one think that I remind you that some sections of the national and international guidelines changed in some extent from time to time due to the nature of the disease and others issues. As such, this may be the limitation for this as well as for other studies.

2. It is indicated that observations were done, but it is difficult to see the findings.

Response: The health facilities observed by incorporating different domains. The observational checklist used for this study was standard checklist, which developed by WHO. The observational finding reported in the result section with heading “Observational findings regarding the practice of precautionary measures for COVID-19 pandemic at the facility level” and “Table 5”. So, please refer that section.

3. If this paper should be consider of general interest we need to know more about the local context: the population, geography / transportation, economy, language / communication channels, education and degree of Covid-19 exposure.

Response: I thank for this comment. As this study was conducted among health professionals, and I incorporating some of the information that you stated in different sections (method: study setting, result: socio-demographic characteristics, et.) of the paper. 

4. A number of technical / formal details need to be corrected. The English needs a total brush-up, a thorough proof reading should be done and abbreviations are not always introduced / used correctly (ODK,AMU,IPC?).

Response: I thanks for this comment. The whole section of the paper is copy edited, and the abbreviation inconsistencies corrected.

ODK: first introduced and used in the data collection section as “Open Data Kit”

AMU: used in the data management and processing section and stated as “Arba Minch University” in the revised version.

IPC: Used in the “Table 5” and stated as “Infection Prevention and Control” in the revised version. 

5. Lastly: in implementation research the dissemination of the results to similar settings and feed-back to the study subjects are essential factors. A paragraph on this should be added. There is room for shortening and reduction of tables to make space for the text I am missing.

Response: I thanks for this suggestion. I incorporated this idea in the last section of the discussion part.

Reviewer #2:

1. It appears that this has been previously published and therefore does not qualify for publication with PLOS ONE.

Response: This article is not published elsewhere rather than the preprint based on the request of journal office which recommended by WHO on COVID-19 researches. I included the DOI of the preprint in medRxiv in the revised version as additional information based on the “journal policy” on preprint. 

2. In terms of the technical content of your paper I find the initial analysis good. However some key issues not dealt with in your paper include the failure to acknowledge that you've interviewed only healthcare professions who are likely to engage in many of the practices you identify as standard practice--even without COVID-19.

Response: I thanks for this comment. Our study assessed the practice of precautionary measures for COVID-19 among health professionals by using interview and observation during the era of COVID-19. There is standard for each practice in the health care system. But, our study was focused only on precautionary measures for COVID-19 even if some practice are similar with previously infection prevention practice which was already incorporated in the health care system. Therefore, we acknowledged that. 

3. You did find that only 68.9% practice some of these standard preventative measures--but there isn't a great deal of discussion as to why these attitudes exist.

Response: I thanks for this comment. In our study, 68.9% had wash hands continuously with water, soap, or use hand sanitizer (which is one component of preventive measures) as you indicated. We incorporated different reasons why this attitude exist, and we compared and contrast with previous studies. This information indicated in the least part of the paragraph that this idea is stated.

4. This reads a bit more like an accounting of healthcare workers perceptions rather than and explanation of why which would have been a bit more useful.

Response: Our paper assessed healthcare workers practice of precautionary measures for COVID-19 as indicated in the finding rather than the perceptions. We also supplemented with observational findings. 

5. Also your multivariate model looks like an attempt to throw everything in and see what ends up being significant. There is not theoretical framework or justification for why the independent variables were chosen and since this is not a grounded theory study I would expect the authors to use some kind of theoretical model to justify their inclusion of these variables. This also leads to a week discussion of these variables and the mechanisms through which they may be influencing the dependent variable (use of good precautionary practices).

Response: I thanks for this comment. We used different assumptions (variables that included in previous studies, the variable that fulfil the assumption of the model, etc.) to include variables to the multivariable model even if theoretical framework not used. I readied some articles that published recently used theoretical model. I crosschecked the variables that used by this study and all the variables that we included in our model were used by those studies. 

With Best Regards!

Abera Mersha Mamo (BScN, MScN, Assistant Professor in Maternity and Neonatal Nursing),

School of Nursing, College of Medicine and Health Sciences,

Arba Minch University, Arba Minch, Ethiopia

Tel: +251(0)-910-389-538/+251(0)-961-413-332 (Ethiopia)

Email: mershaabera@gmail.com/

mersha.abera@yahoo.com/

abera.mersha@amu.edu.et

---

## [Decision Letter · Decision Letter 1]

24 Feb 2021

Health professionals practice and associated factors towards precautionary measures for COVID-19 pandemic in public health facilities of Gamo zone, southern Ethiopia: a cross-sectional study

PONE-D-20-27501R1

Dear Dr. Mersha,

We’re pleased to inform you that your manuscript has been judged scientifically suitable for publication and will be formally accepted for publication once it meets all outstanding technical requirements.

Kind regards,

Muhammad Adrish

Academic Editor

PLOS ONE

Additional Editor Comments (optional):

All queries have been addressed.

Reviewers' comments:

Reviewer's Responses to Questions

**Comments to the Author**

1. If the authors have adequately addressed your comments raised in a previous round of review and you feel that this manuscript is now acceptable for publication, you may indicate that here to bypass the “Comments to the Author” section, enter your conflict of interest statement in the “Confidential to Editor” section, and submit your "Accept" recommendation.

Reviewer #1: All comments have been addressed

2. Is the manuscript technically sound, and do the data support the conclusions?

Reviewer #1: Yes

3. Has the statistical analysis been performed appropriately and rigorously? 

Reviewer #1: I Don't Know

4. Have the authors made all data underlying the findings in their manuscript fully available?

Reviewer #1: Yes

5. Is the manuscript presented in an intelligible fashion and written in standard English?

Reviewer #1: Yes

6. Review Comments to the Author

Reviewer #1: The main comments are satisfactorily met by the authors. No further comments

7. PLOS authors have the option to publish the peer review history of their article (what does this mean?). If published, this will include your full peer review and any attached files.

Reviewer #1: **Yes: **Gunnar Aksel Bjune

---

## [Editor Report · Acceptance letter]

1 Mar 2021

PONE-D-20-27501R1 

Health professionals practice and associated factors towards precautionary measures for COVID-19 pandemic in public health facilities of Gamo zone, southern Ethiopia: a cross-sectional study 

Dear Dr. Mersha:

I'm pleased to inform you that your manuscript has been deemed suitable for publication in PLOS ONE. Congratulations! Your manuscript is now with our production department. 

Kind regards, 

on behalf of

Dr. Muhammad Adrish 

Academic Editor

PLOS ONE